# Probing Representations of Numbers in Vision and Language Models

**Ivana Kajić**
DeepMind
kivana@deempind.com

**Aida Nematzadeh**
DeepMind
nematzadeh@deempind.com

## Abstract

The ability to represent and reason about numbers in different contexts is an important aspect of human and animal cognition. Literature in numerical cognition posits the existence of two number representation systems: one for representing small, exact numbers, which is largely based on visual processing, and another system for representing larger, approximate quantities. In this work, we investigate number sense in vision and language models by examining learned representations and asking: What is the structure of the space representing numbers? Which modality contributes mostly to the representation of a number? While our analyses reveal that small numbers are processed differently from large numbers, as in biological systems, we also found a strong linguistic contribution in the structure of number representations in vision and language models, highlighting a difference between representations in biology and artificial systems.

## 1 Introduction

Whether it is foraging for food in novel environments, or counting slices of cake at a birthday party, humans and animals frequently demonstrate various aspects of numerical competence. Reasoning about quantities is a fundamental cognitive feature, and it plays an important role in survival and successful reproduction. Literature in numerical cognition points to two systems of number representation; an approximate system that supports intuitive reasoning about numerical magnitudes, often been referred to as *number sense* (Dehaene, 1997; Lipton & Spelke, 2003), as well as the system in charge of rapid, confident and accurate discrimination of small numerosities known as *subitizing* (Kaufman et al., 1949). Thus, natural intelligence is endowed with neural mechanisms that support emergence of number competence as a by-product of exposure to natural visual stimuli, without any explicit training for numerosity estimation (Nieder, 2020).

In contrast, the majority of existing work in artificial intelligence has focused on more advanced aspects of numerical competence, such as counting (Zhang et al., 2018; Mandler & Shebo, 1982; Trott et al., 2018), arithmetic or quantitative reasoning (Geva et al., 2020; Drori et al., 2022; Lewkowycz et al., 2022). While these are important components of analytical skill, they are either gradually acquired during development or via formal education, and are thus associated with various higher-level cognitive functions. Fewer studies have investigated how more fundamental notions of numerical competence, such as those that have been documented in naïve animals or prelinguistic infants, develop in artificial systems (Creatore et al., 2021; Testolin et al., 2020). In this work, we ask whether contemporary deep neural networks trained on large amounts of static image-text data learn representations that are functionally comparable to those underlying number processing in biological cognition. We study whether distinct number representations can be observed for small versus large numerosities and how this depends on the specific modality. Our findings suggest that number representations in vision and language models are structured in a way that is consistent with the

4th Workshop on Shared Visual Representations in Human and Machine Visual Intelligence (SVRHM) at the Neural Information Processing Systems (NeurIPS) conference 2022. New Orleans.

two-system theory in numerical cognition. When comparing modalities, we found that language contributes more than vision towards accurate number representation. We argue that biases in the data used to train these models might explain some of our findings.

## 2 Related Work

Numeracy-related research in artificial intelligence and machine learning spans a spectrum of motivations. On the one hand, there is a strong application-driven incentive to improve performance and the quality of representations on tasks requiring numerical skills (*e.g.,* arithmetic, magnitude comparison or numerical common sense, among others) which has been lagging behind other NLP benchmarks (Thawani et al., 2021; Wallace et al., 2019; Parcalabescu et al., 2021; Lin et al., 2020). Helpful approaches improving performance on such tasks include using curated synthetic data (Geva et al., 2020; Zhang et al., 2015), or heuristics such as counting-specific model components and processes (Zhang et al., 2018; Trott et al., 2018; Ranjan et al., 2021).

On the other hand, understanding how the more abstract concept of a number is related to visually perceived numerosity is associated with the grounding problem studied in artificial intelligence and cognitive science (Harnad, 2003). Understanding the representation of numbers, and factors that affect it, offers the potential of enriching such representations in artificial systems by informed inductive biases. Creatore et al. (2021) show that basic neural networks can develop internal representations that support qualitatively different numerosity perceptions systems, akin to number sense and subitizing, though with some differences compared to human processes. Wallace et al. (2019) find that token embeddings learned from text can accurately encode magnitudes for numbers within the training range, while failing to extrapolate to numbers outside the range. While affirming that linguistic data contains a significant amount of information about numbers, it is unclear how these representations relate to biological ones, where small numerosities are perceived differently from larger ones.

As well, the existing work in ML related to numeracy has predominantly been focused on specific benchmark performance (Parcalabescu et al., 2021; Zhang et al., 2015; Wallace et al., 2019), and less so on fine-grained analysis of learned number representation and their extension to visual data. In this work, we scrutinize these representations in the context of what is known about number representation in biological systems.

## 3 Methods: Models and Datasets

Transformer-based neural network models (Vaswani et al., 2017) have become a de facto standard modelling choice in NLP since their inception, and the extension to support the visual modality has opened doors to study a new space of problems and multimodal interactions (Du et al., 2022; Hendricks et al., 2021; Bugliarello et al., 2021). Here, to characterize representations of numbers, we use the VILBERT (Lu et al., 2019) architecture, a vision and language model that extends the BERT (Devlin et al., 2018) architecture to support processing of visual and text inputs. It processes inputs via two separate, parallel streams, which are subsequently combined via co-attentional transformer layers. Text is presented as a sequence of tokens, while image is serialized as a sequence of region-of-interest features extracted from a separate convolutional neural network (Ren et al., 2015).

Most vision and language models are pre-trained on pretext tasks with multimodal data, analogous to the training process used with text-based transformer models. Then, models are further fine-tuned on a specific transfer task such as visual question answering (VQA), visual commonsense reasoning, referring expressions, or caption-based image retrieval by learning a task-specific decoding head. The linguistic stream of the VILBERT model is initialized with BERT weights, and the model is pretrained on 3.3M image-text pairs from Conceptual Captions (CC; Sharma et al., 2018).

Here, we also study the model that has been fine-tuned on VQA datasets which include explicit number-related questions. In VQA, the task is to answer a question about a given image. We investigate representations in models that have been fine-tuned on two widely used VQA datasets: VQAv2 (Goyal et al., 2017), and Visual Genome (VG; Krishna et al., 2017). While the original VILBERT implementation uses a classifier as a decoding head, we use auto-regeressive token decoder, which is more flexible as it does not require a priori specification of the number of output classes. In VQAv2 there are 10 human responses for each image-question pair, while there is just one

response in VG. In our analyses we focus on number-related questions (*i.e.,* those that start with *How many...* or *What number...*), and use the existing data splits. In total, such questions represent about 11.5% and 8% of all questions in VQAv2 and VG, respectively.

## 4 Results

### 4.1 Analysis: Number Representation Similarity

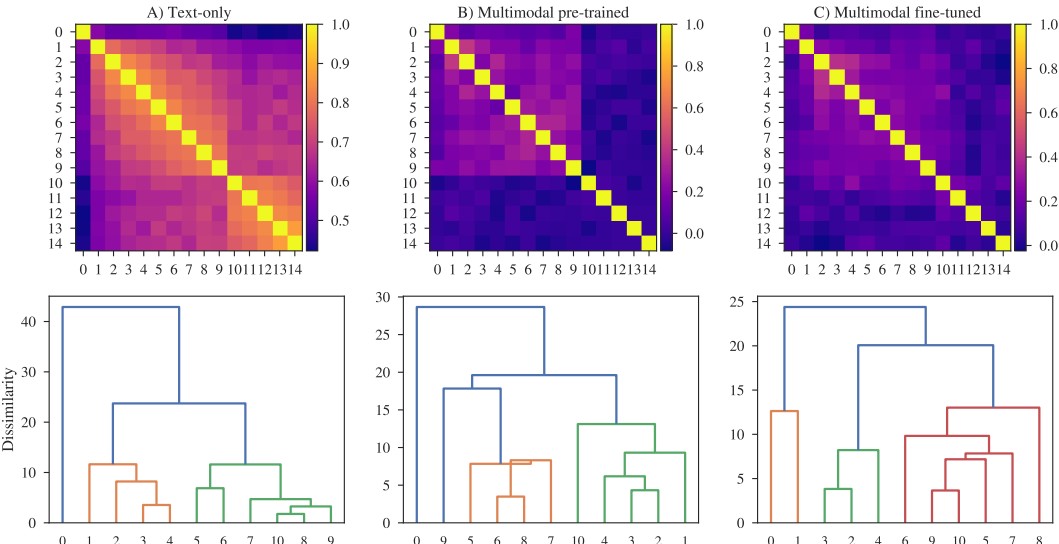

Figure 1: Visualizing similarities between number embeddings. Top: Pairwise cosine similarities between token embeddings for number tokens 0 to 14. Bottom: Dendrograms showing the hierarchy of number clusters based on similarities between embeddings.

First, we look at differences in number representations between text-only models and multimodal models. Rather than looking at aggregate benchmark performance on number-related tasks (Parcalabescu et al., 2021; Wallace et al., 2019), we examine the relationship between learned number representations. Specifically, we are interested in whether learned representations capture any structure reflecting the order of numbers at the qualitative level, and the change in structure during the pre-train/fine-tune process.

We extract learned token embeddings for numbers 0 to 14 from three models: BERT (text-only), VILBERT pre-trained on CC (multimodal), and pre-trained VILBERT fine-tuned on VG (multimodal). For two-digit numbers, we only consider a single token (*i.e.,* "14" instead of "1" and "4"). Figure 1 (top) shows pairwise cosine similarities between extracted token embeddings. Text-only embeddings, shown in Figure 1 (A), display a pattern of 3 visually distinct clusters: one for the token 0, one for numbers 1-9, and one for numbers 10 and larger. Multimodal pre-training appears to distort that pattern, especially as numbers from 10 to 14 become less similar to any other numbers. This occurs because captions in CC contain only numbers from 0 to 9. As well, CC has a peculiar distribution of numbers, with 3 and 4 being the most frequent, possibly due to common occurrences of "3D" and "4K" tokens in the dataset.

To highlight the similarity between individual representations, we plot results of hierarchical clustering of representations of numbers from 0 to 10 in the bottom of Figure 1.[1] Clustering was performed on 2D PCA-projected representations of token embeddings, using the centroid method and the Euclidean metric for calculating the distance between clusters. The algorithm starts by treating each token as an individual cluster, and proceeds to iteratively merge least dissimilar clusters. In all cases, we observe that some of the first clusters formed from singletons are those for subsequent numbers (*e.g.,* some of them are (3, 4) and (5, 6) for text-only representations; (2, 3) for multimodal pre-trained representations; and (0, 1) and (2, 3) for fine-tuned multimodal representations). We observe further

---

[1]This is the range present in all conditions we examined.

interpretable groupings in the case of BERT, as 2 is merged with (3, 4), 1 with (2, (3, 4)), 7 with (9, (10, 8))) etc. For fine-tuned representation a cluster is formed for (2, 3) and 4, and another one for all larger numbers. In general, we also observe anti-patterns (*i.e.,* 10 merging with 8 for BERT; or 10 with small numbers for multimodal pre-trained).

Restricting the analysis to the following three clusters: 0, small numbers within subitizing range (*i.e.,* 1–4), and larger numbers outside the subitizing range (*i.e.,* 5–10), which we take to be the gold standard reflective of number representation structure in humans and animals, allows us to evaluate cluster assignments observed in hierarchical clustering. Specifically, we consider cluster assignments at points where the distance cut-off value defines three clusters for each dendrogram in Figure 1. To compute the $F_1$ score, we follow the approach outlined in Schütze et al. (2008, Section 16.3): $F_1$ is computed as $\frac{2\,P\,R}{(P+R)}$, where $P$ is precision defined over true/false positive (TP/FP) decisions as $\frac{TP}{TP+FP}$, and recall $R$ as $\frac{TP}{TP+FN}$, where FN stands for false negative decisions. When calculating TPs, FPs and FNs we consider a decision over a pair of numbers over all possible combinations of number pairs: a TP is is when two numbers from the same gold standard cluster end up clustered together, a FP assigns the same cluster to two numbers from two different gold standard clusters, and a FN assigns two numbers from the same gold standard cluster to two different clusters.

The highest $F_1$ score is observed for text-only representations ($F_1 = 1.00$), followed by multimodal fine-tuned on VG ($F_1 = 0.90$), multimodal-pretrained ($F_1 = 0.78$), and multimodal fine-tuned on VQAv2 ($F_1 = 0.53$). This leads us to conclude that text-only number representations are structured in a way that is most similar to the structure of number representations in humans and animals.

## 4.2    Influence of Modality and Numerosity in Number Representation

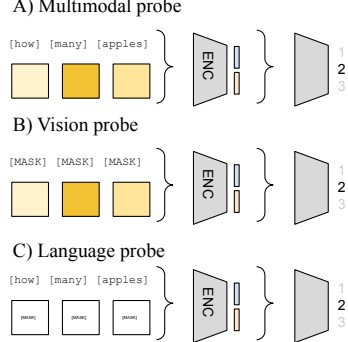

Figure 2: Probes for multimodal, visual and language number representations.

| Dataset | Probe | (1, 4) | (5, 10) | (1, 10) |
|---------|-------|--------|---------|---------|
| VQAv2 | Vis | 44.22 | 23.71 | 37.36 |
| VQAv2 | Lang | 46.59 | **31.26** | 38.75 |
| VQAv2 | [Lang, Vis] | **55.09** | 29.04 | **46.39** |
| VG | Vis | 56.72 | 31.81 | 52.66 |
| VG | Lang | 58.57 | **37.35** | 54.46 |
| VG | [Lang, Vis] | **69.10** | 31.61 | **64.17** |
| Random | n/a | 25.27 | 16.12 | 10.46 |

Table 1: Classification accuracy on test sets for number labels from pooled feature representations.

In biological systems, the processing of items within the subitizing range is attributed to the visual system, while larger numbers are assumed to be processed in a different way (Kaufman et al., 1949). In the previous analysis we found text-based number representations to be the most interpretable. However, it is unclear which modality contributes the most to the representation of a number in multimodal models, and whether this depends on the number range. In this section, we design a probe to answer that question, inspired by similar work in the domain (Lin et al., 2020; Wallace et al., 2019; Parcalabescu et al., 2021).

We train probes to predict numerals based on features extracted from different modalities: multimodal (concatenated visual and linguistic features), visual, and linguistic. Features are extracted from a fine-tuned model as pooled representations of an input question (text) or image (vision) from the 'CLS' (for text) or 'IMG' (for images) tokens at the encoder output during the forward pass on a dataset. In other words, for each (question, image) tuple from a VQA dataset we get two feature vectors. For the vision probe, used to examine the contribution of visual modality in representing numbers, we entirely mask the input question; for the language probe, we entirely mask the visual input to the model. Figure 2 illustrates the probing process. The probe is trained to minimize the cross-entropy loss when predicting the corresponding numeric label—the answer associated with the (question, image) tuple. By ablating one modality in this way, we can study the contribution of the other modality in predicting numbers. Each probe is trained on features from a train split of a corresponding dataset, and tested on the val split of the same dataset. As well, separate probes

are trained for different number ranges. Further details on training and evaluation are provided in Appendix A.1.

The probing results are shown in Table 1. In most cases, multimodal features are best at encoding numbers compared to features from a single modality. They are better at encoding smaller numbers (*i.e.,* 1–4) than the larger numbers (*i.e.,* 5–10). As for individual modalities, linguistic features appear not only to be better at encoding numbers than the visual features, but also better than multimodal features for larger numbers. The finding that language is more informative of small numbers than vision is an interesting difference between number representation in biology and deep neural networks, as animals and pre-linguistic infants are able to subitize without having developed or acquired language. This could perhaps be due to vision being sufficiently informative, so that linguistic input does not provide further benefit in this process. The reason why it might be somewhat easier to predict the number given a masked image, than to predict the number given masked text is that the question type *How many...* or *What number...* is more informative of the space of possible answers than the image itself. Given an image, without any text, the space of possible answers is more diverse (*i.e.,* yes/no answers, number, color, nouns, verbs etc).

## 5  General Discussion

The ability to represent and reason about numerical quantities has been extensively studied in human and animal cognition. Human brains, as well as brains of other animals, are equipped with a form of rudimentary number sense essential for survival and reproduction (Dehaene, 1997; Nieder, 2020). In this work, we investigated whether contemporary neural networks processing visual and linguistic inputs develop a notion of a number that is comparable to that observed in biological systems. Namely, we investigated how are numbers represented, and whether small numerosities in the subitizing range (*i.e.,* 1–4 items) are processed differently from large numerosities.

First, we found interpretable structure among number representations—in some instances, representations between subsequent numbers were more similar compared to representations between non-subsequent numbers. When we coarsely clustered number representations into groups based on how numbers are represented and processed in biological systems (small numbers in the subitizing range vs. numbers outside that range), we surprisingly observed a perfect score for number representations coming from a model that has only been trained on text (BERT). We speculate that number ordering, as observed during cluster merge process, as well as grouping of small vs larger numbers could be due to the statistical distribution of numbers in the training text. That is, pairs of numbers such as (1, 2) or (3, 4) are more likely to occur than (1, 5) or (2, 4). In addition, the Newcomb-Benford law (Newcomb, 1881; Benford, 1938), stating that leading digits are likely to be small, might imply better representation of numbers within the subitizing range in real-world data, which could explain some of the patterns we observe. It is worth noting that the distribution of digits in multi-modal data did not adhere to that law.

Second, we examined to what extent individual modalities in vision and language models contribute to the representation of a number. We designed a probe that ablated one modality and learned to predict numbers based on the inputs to the other, non-ablated modality. While multimodal features were best at predicting the number overall, and especially small numbers in the subitizing range, linguistic features were better at this task than visual features. We found this result surprising in light of the fact that humans and non-human animals develop numerical competence through exposure to natural visual stimuli. The higher accuracy on smaller number ranges for all modalities is also likely to be explained by better representation of small numbers in training data, which is the case for both datasets. Small numbers in the range 1–4 account for 83.6% (VQAv2) and 93.1% (VG) of probe training data, with remaining numbers being 5–10 (see Fig. A.2 for more details). As a reference, the Newcom-Benford law predicts that numbers 1–4 would account for approximately 70% of data. We consider it an open question as to why linguistic features appear better than multimodal features for the representation of larger numbers (*i.e.,* 5–10).

In future work, we would like to better examine the role of pre-training, robustness and generality of learned number representations. Since vision and language models are known to latch onto surface-level correlations in the data in VQA  (Goyal et al., 2017; Agrawal et al., 2016), it is unclear how transferable learned number representations are. As well, due to distributional statistics of numerical data in these datasets, it is difficult to discern whether subitizing-like patterns we observe are simply

due to better representation of small numbers, or are indicative of an emergent phenomenon with distinctive cognitive and behavioural characteristics.[2] We postulate that systematic assessments, similar to those used in cognitive science and psychology, might help to accurately characterize the role of different factors contributing to number sense in artificial systems.

# 6 Acknowledgments

The authors would like to thank anonymous reviewers, as well as their colleagues at DeepMind for advice and feedback that helped improve this manuscript. They also thank Emanuele Bugliarello for a detailed review of the paper draft.

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

# A   Appendix

## A.1   Technical Details for Probing Experiments

Our probe is an MLP with 2 hidden layers, with 100 units in each, and a linear layer at the output (10 units). Output of each unit in each layer is passed through a ReLU non-linearity. Training labels are numbers from 1–10, encoded as one-hot vectors. We use cross-entropy loss at the output, which is minimized using Adam optimizer (Kingma & Ba, 2014) with a learning rate of 0.001. We train the probe for 50K steps using feature vectors (*i.e.,* 'CLS' or 'IMG' tokens) extracted from the forward pass of a training split of VQA dataset through the VILBERT model, and evaluate on the val split of the same dataset. We only extract features from (image, question) pairs where the question starts with "How many" or "What number". We normalize answers so that "1." and "1" is the same answer. For VQAv2, we use the most frequent response among ten responses as the ground truth answer for the probe.

For VILBERT pre-training and fine-tuning we use 16 TPUv3s, while for evaluation (collecting pooled features) we use 1 GPU. GPUs are either Tesla V100 or P100.

## A.2   Distribution of Number Answers in Probe Datasets

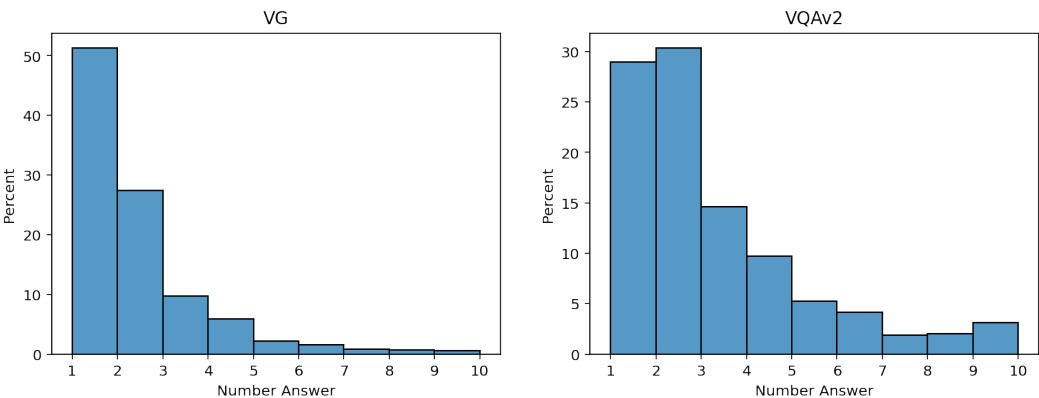

Figure 3: Normalized distribution of number answers in the training split of VQAv2 and VG datasets used for training of Probe in Sec. 4.2.

