# OpenReview forum: "Probing Representations of Numbers in Vision and Language Models"
_NeurIPS.cc/2022/Workshop/SVRHM — SVRHM Poster_

### Official Review · Reviewer_HZcu · 2022-10-12
**Probing Representations of Numbers in Vision and Language Models**

**Rating:** 7
**Confidence:** 3

**Review:**

Summary:
 This work presents an analysis of the number representation of two modalities, visual and language by masking one or the other, and their combination. Besides, it explains the two-system theory to process biology cognition of numbers and compares it to the clusters found based on the distance of number representations. The results indicate the multi-modal models represent much better the biological system and counterintuitively that language models represent them better than visual models.


Pros

Very well explained paper and interesting idea that reflects the intersection of the ML and psychology fields. It presents an interesting start to get closer to provide ML models a sense of how realy cognition models approach to number processing.

Cons

Although the comparison seems to be fair, the same multi-modal model using masks for certain modality, it is not clear how the output of the language modality can be proved more similar to the biology-based approach. It sounds like any label could be correct. It could suggest that the results are biased due to an imbalance dataset. Authors should also provide distribution of the labels.

To show more evidences on the different contribution on number representation from ML compared biology systems, authors are encourage to test other other VQA models and/or test independent multi-modal, language and visual models.

Authors should provide more information on what are the regions-of-interest features used as the input sequence for the visual modality. If authors consider analyzing independent visual model, perhaps providing one more image reference (i.e. what object to count) could help provide a more robust expalanation on why the visual modality results.

If it is a metric slightly different as the usual, F1 Score calculation should be explained, additionally to mention its reference.
The authors mention the clustering over the 2D projection of the representation embedding. It would be great to know what percentage of the whole representation is 2D, and mention the size of the representation in the modalities.

Explain better the auto-regressive variation added to VilBERT and how the output is transformed/mapped to final labels.

Highlight or bold the most important values y table.

---

### Official Review · Reviewer_X5uU · 2022-10-13
**Nice initial analysis but further work required**

**Rating:** 4
**Confidence:** 3

**Review:**

The authors present a nice comparison of using ViLBERT based on text and vision data and aim to identify to what degree ANNs learn two number systems similar to those found in humans, identifying a difference between the subitizing range (1-4) as compared to larges numbers. I think this is an interesting question that deserves to be studied. The manuscript is overall well written.

I think this work aims at addressing an interesting question but I'm not sure it is quite ready to be presented. The analyses appear to be rather superficial and could be motivated better. In addition, to me it is unclear why this specific architecture was used to make general statements about whether representations in vision and language models are similar to those found in humans. Why not use separate language models and vision models for addressing this question? Why is a transformer used for static embeddings when semantic embeddings are better suited for that purpose, and what would we learn if a similar representation was learned when we are not dealing with a transformer? And is it sufficient to show that numbers 1-4 are represented more similarly to each other? I am not an expert in number representations but I believe that humans are very good at guessing the number of items from just seeing them when there are 4 or fewer. I just think that more evidence is required than showing a separation between numbers 1-4 and 5+ to assume that two systems emerge naturally in ANNs.

I also couldn't understand what exactly was compared for the analyses shown in Figure 1: was it the average representational similarity of all pairs of token representations, or the representational similarity of all pairs of average token representations (for all numbers), or something else? I believe more details about the methodological steps would help. Also, why did the authors focus on the token embedding only? Why does Figure 1 (bottom) only include 10 numbers? (there is a footnote that appear a bit opaque).

Overall I think this is a very promising starting point that is well worth continuing! I believe the authors should consider these questions when moving forward.

---

### Official Review · Reviewer_cp6J · 2022-10-16

**Rating:** 6
**Confidence:** 3

**Review:**

Summary:
This work studies representations of small and large numbers in vision-and-language models. Models trained on visual-question answering tasks related to numbers structure representations consistent with two-system theory, and language contributes more to accurate number representation than vision. It is surprising in particular that clusters obtained from BERT representations obtain a perfect F1 score.

Concerns:
- In the clustering experiments, fine-tuning the pre-trained model on VG increases the F1 score, while fine-tuning on VQAv2 significantly decreases it. Why is this happening?
- Are these results merely reflecting the statistics of the training data, or is there some distinct phenomenon related to learned number representations taking place? It would be useful to see statistics for token appearances in pre-training and fine-tuning datasets.
- The MLP used for the probe in the ablation test is a small 2-layer network. Is there some motivation for using this architecture over a bigger one? In particular, would the results remain consistent if a much larger network were fit to the same task?

---

### Official Review · Reviewer_SGR1 · 2022-10-17
**Well-executed experiments, but with unclear implications about the motivating question**

**Rating:** 6
**Confidence:** 3

**Review:**

The authors here aim to investigate whether neural networks represent small numbers differently than larger numbers, just as humans do. In order to do so, they investigate the representational similarity between representations of different numbers and how they are clustered together. In aiming to understand how distinctly different numbers are represented in neural networks, the authors' analyses are well executed. In particular, their finding that the clusters of the text-based representations align perfectly with the groups 0, 1-4, and 5-9 is intriguing. However, this analysis, in my view, does not imply that "small numbers are processed differently than large numbers, as in biological systems". While there are such differences in the networks, I believe the authors should make clearer how the analyses they're conducting are related to the distinction between subitizing and non-subitizing. While the increased task performance for small numbers may suggest that networks are better at tasks involving those numbers, it is unclear, for example, whether we should consider the Transformers' output to be the immediate response of a human (thus only being able to precisely count numbers in the subitizing range) or the response after an amount of reflection. Since subitization appears intimately connected with processing time, it may also be of interest to explicitly model that processing time. The authors' submission is, however, contributing to this debate and providing an initial analysis. I therefore see this as a paper on the threshold.

Minor comments:
l. 14: “... humans and animals demonstrate various aspects of numerical competence on a daily basis”
l. 68 maybe rather “... their extension to visual data”?
The definition of the F1 score here is a bit unclear and you may want to elaborate on how exactly it is defined in this case.
l. 160-162: Does this necessarily suggest, though, that in humans language is less informative? It may just be that vision is also sufficiently informative that we don’t need language.
l. 157 “In fact, training on the full range of 10 numbers reduces the accuracy.” This is not apparent from the table necessarily, right? It is, for instance, still possible that the accuracy on (1,4) is as high as previously, as the table only specifies the aggregate performance on (1,10).